# NMC3D: Non-Overlapping Multi-Camera Calibration Based on Sparse 3D Map

**DOI:** 10.3390/s24165228

**Published:** 2024-08-13

**Authors:** Changshuai Dai, Ting Han, Yang Luo, Mengyi Wang, Guorong Cai, Jinhe Su, Zheng Gong, Niansheng Liu

**Affiliations:** 1School of Computer Engineering, Jimei University, Xiamen 361021, China; dcs740328997@gmail.com (C.D.); luoy1014116@gmail.com (Y.L.); wangmengyi57@gmail.com (M.W.); guorongcai.jmu@gmail.com (G.C.); sujh@jmu.edu.cn (J.S.); gongzheng@stu.xmu.edu.cn (Z.G.); 2School of Geospatial Engineering and Science, Sun Yat-Sen University, Zhuhai 519082, China; ting.devin.han@gmail.com

**Keywords:** calibration, SLAM, multi-camera, feature selection

## Abstract

With the advancement of computer vision and sensor technologies, many multi-camera systems are being developed for the control, planning, and other functionalities of unmanned systems or robots. The calibration of multi-camera systems determines the accuracy of their operation. However, calibration of multi-camera systems without overlapping parts is inaccurate. Furthermore, the potential of feature matching points and their spatial extent in calculating the extrinsic parameters of multi-camera systems has not yet been fully realized. To this end, we propose a multi-camera calibration algorithm to solve the problem of the high-precision calibration of multi-camera systems without overlapping parts. The calibration of multi-camera systems is simplified to the problem of solving the transformation relationship of extrinsic parameters using a map constructed by multiple cameras. Firstly, the calibration environment map is constructed by running the SLAM algorithm separately for each camera in the multi-camera system in closed-loop motion. Secondly, uniformly distributed matching points are selected among the similar feature points between the maps. Then, these matching points are used to solve the transformation relationship between the multi-camera external parameters. Finally, the reprojection error is minimized to optimize the extrinsic parameter transformation relationship. We conduct comprehensive experiments in multiple scenarios and provide results of the extrinsic parameters for multiple cameras. The results demonstrate that the proposed method accurately calibrates the extrinsic parameters for multiple cameras, even under conditions where the main camera and auxiliary cameras rotate 180°.

## 1. Introduction

Multi-camera systems are employed to capture various perspectives using multiple cameras, providing the advantage of a wide field of view and comprehensive information capture. Furthermore, the utilization of multiple views serves to compensate for deficiencies, such as the inability to measure distances directly. Multi-camera systems possess these advantages, making them widely used in robot obstacle avoidance [1,2,3], path planning [4,5,6], visual navigation [7,8,9], and 3D reconstruction [10,11,12]. In the field of robotics, multiple cameras fixed on robot components form a multi-camera system. This system is employed to help the robot more effectively perceive its surrounding environment and make decisions. However, a fundamental prerequisite for implementing a multi-camera system is the accurate knowledge of the relative positions of the cameras, meaning the extrinsic of the multi-cameras.

In existing methods, a specific calibration site is usually set up, and a calibration plate is placed at an overlapping position of the camera fields of view. The calibration of the multi-camera system is then performed by selecting a feature matching point on the calibration plate in the overlapping portion. This method is prone to failure when there are no or few overlapping parts between the multi-camera systems, which in turn leads to an inability to complete the calibration of the multi-camera systems. Once the feature matching process is complete, existing methods randomly select the matching points for subsequent calibration. These approaches fail to fully utilize the nature of the matching points, making it difficult to achieve high-precision calibration of the extrinsic of the multi-camera system. Therefore, it is summarized that there are still two major problems in calibrating using existing multi-camera calibration methods: The first challenge is to determine the optimal calibration settings for a multi-camera system when there is no overlap between multiple cameras. The second is how to make full use of the matching points for calibration when obtaining feature matches.

The absence of overlap between multi-camera systems introduces greater difficulty in using site-specific calibration plates. In the case of non-overlapping parts, the extrinsic calibration of multiple cameras cannot be accomplished because the feature points of the calibration plate cannot be extracted from multiple cameras at the same time. Some researchers [13,14] have proposed the use of plane mirrors to project the image of the calibration plate onto the overlapping portions of multiple cameras. The method of using plane mirrors enables the simultaneous extraction of the feature points of the plate, thus completing the calibration. Although the method of using plane mirrors allows for the calibration of multiple camera extrinsic parameters, it also significantly increases the labor cost and reduces the portability of the algorithm. The method of using plane mirrors necessitates the utilization of a specific calibration site and plane mirror in order to achieve the calibration of the extrinsic. To address the above problems, we calibrate the extrinsic of the multi-camera system with no overlapping parts by using natural environments instead of specific calibration sites and plane mirrors. Firstly, the closed-loop motion is carried out by the multi-camera system to ensure that each camera can capture similar images. These images are then used to obtain the matching features, which are subsequently employed to achieve the calibration when there is no overlapping region between the multi-cameras. The capture of images of natural scenes in lieu of specific calibration sites and plane mirrors not only reduces the labor costs associated with the process but also enhances the practicality of calibration.

In the process of acquiring feature match points for calibrating a multi-camera system, existing methods randomly select points among the feature matches to complete the subsequent calibration [15]. The method for randomly selecting matching points fails to take full advantage of the nature of the feature match points and the extent of the distribution. To address this problem, we propose a distance-based feature matching point selection strategy to complete the subsequent calibration of the multi-camera system. First, we set the thresholds of far-distance and near-distance points in the feature matching points. Then, we screen out the far-distance point set, medium-distance point set, and near-distance point set in the feature matching point set. Finally, we select the feature matching points in each point set and complete the subsequent calibration of the multi-camera system by choosing the feature matching points with different distances and uniform distribution. By selecting feature matching points with different distances from each other, it greatly influences the computation of rotation and translation vectors in the extrinsic parameters of multi-camera systems. Similarly, feature matching points with homogeneous distributions is able to significantly improve the accuracy of computing multi-camera extrinsic parameters. The main contributions of this paper are as follows:We propose a method for constructing 3D maps of natural environments by adding fixed constraints to calibrate multi-camera systems without overlapping parts.We propose a distance-based feature matching point selection strategy to improve the accuracy and robustness of calibration.The efficacy of the method is demonstrated by its capacity to facilitate rapid deployment and minimal latency in multi-camera systems.

## 2. Related Work

In this section, we review some multi-camera calibration methods. The calibration methods are categorized into artificial target-based methods and targetless methods, based on whether artificial references are used during the calibration.

### 2.1. Target-Based Multi-Camera Calibration Methods

The process of multi-camera calibration utilizes 3D objects as a reference to provide accurate 3D geometry. Agrawal et al. [16] proposed a spherical occlusion contour-based method for camera calibration. In the majority of cases, spheres were represented in the image as ellipses and were closely related to the camera parameters. The algorithm employed three spheres for the simultaneous calibration of the camera parameters. The process of multi-camera calibration was typically performed using 2D flat objects, with specially designed flat patterns incorporated for the purpose of calibration. An algorithm for multi-camera calibration using a pattern plane with a known reference point was proposed by Ueshiba et al. [17]. The algorithm divided the single responsivity matrix between the pattern and image planes into camera projection and plane parameters. Subsequently, the projection and planar parameter matrices were transformed into a coordinate system, utilizing metric information for the purpose of calibrating the multi-camera. The application of multi-camera calibration using 1D objects was demonstrated to establish that the feature points were aligned on a straight line. The work of Zhang et al. was among the earliest to introduce 1D objects into camera calibration [18]. The process of finding a closed solution for a 1D object at a fixed point involved making multiple observations of the object and then optimizing it using nonlinearities. This was done by first establishing the covariance and distance constraints of the marked points and then determining the optimal solution based on these constraints.

The aforementioned artificial target-based multi-camera calibration method was capable of attaining high levels of calibration accuracy when there was a substantial degree of overlap between multiple cameras. However, when there was minimal or no overlap between multiple cameras, such methods were not useful for calibration in this case. A subsequent proposal was made for the use of a calibration method that incorporated a plane mirror in this situation. The objective was to create a single calibration object that could be simultaneously observed by both cameras through the use of a plane mirror. A non-overlapping multi-camera calibration method utilizing mirrors was proposed by Kumar et al. [19]. By maintaining a fixed position for both the camera and the calibration pattern, the mirror was moved to allow the camera to observe the calibration pattern in a variety of positions. In practice, this method was expected to result in a significant economic cost and severely limit the portability of the method.

### 2.2. Targetless Multi-Camera Calibration Methods

The relative attitude parameters between multiple cameras were obtained using motion constraints and estimation in this method. Images captured by one of the cameras, typically designated as the primary camera in most mobile multi-camera devices, undergo processing through an image-building algorithm. This algorithm generates a map based on the captured images, which are then calibrated to align with the feature points in the secondary camera images. An automated multi-camera external parameter calibration method was proposed by Carrera et al. [20]. In this method, a multi-camera device is mounted on a robot and subjected to a series of sequenced motions, such as rotating once a week. The monocular vision SLAM algorithm was also executed with the image sequences captured by the cameras, and the maps were subsequently fused. Finally, the outliers in the multi-camera setup were estimated by matching the image feature points to the map points. In their study, a multi-camera system was calibrated by Heng et al. [21] through the initial construction of a 3D map of the calibration environment. Subsequently, images of the calibration environment were captured using the multi-camera system, and these images were matched to a point cloud map of the calibration environment.

The above method, while it provided a portable solution for calibrating multiple cameras without overlapping portions between them, faced challenges in scenarios with high repetition or weak texture in the calibrated pictures captured by each camera. In such cases, the matching of 2D–3D points was not well accomplished, resulting in decreased calibration accuracy.

## 3. Methodology

In this section, we propose a multi-camera calibration method based on sparse map points, and the general framework of the method is shown in Figure 1. The proposed method included a mapping module and a calibration module as well as an optimization module. Specifically, first in the mapping module, the images captured by each camera were input and the corresponding 3D sparse maps were output by the VSLAM algorithm. Secondly, in the calibration module, the multi-camera extrinsic parameter was calculated by utilizing feature matching points in multiple 3D sparse maps. Finally, in the optimization module, the extrinsic parameters output from the calibration module were used to output the final extrinsic parameters through bundle adjustment [22].

### 3.1. Mapping Module

Inspired by the SLAM framework ORB-SLAM3 [23], multi-sensor data such as monocular, binocular, or RGB-D cameras were used as input. From the input image data, feature points were extracted and matched, the camera pose was estimated, the depth of the feature points was computed, and the 3D map was constructed from the pose and depth. The map-building algorithm comprised three distinct threads: the tracking thread, the mapping thread, and the LoopColsing thread. Firstly, feature extraction was conducted on the input sensor data, and these features were matched to track the movement of the camera. Secondly, the map and camera pose were initialized using triangulation techniques. Moreover, the algorithm estimated the camera’s position by matching feature points with new input frames, which typically included rotations and translations. Then, the map was updated to include new points and was continuously optimized. Finally, a loopback detection module was added to correct the estimated trajectory and improve positioning accuracy.

The mapping module addressed the subsequent calibration operations using the map points of the built environment. The objective was to optimize the built section by adding fixed constraints to it, aiming to improve the calibration accuracy by making the map points obtained through the ORB-SLAM3 algorithm more accurate. The specific method involved using the known 2D distances between corner points on the chessboard board as fixed constraints. By comparing these distances with the calculated distances between the corresponding 3D points, we obtained a difference value. This difference was then incorporated into the projection formula as a factor to enhance its accuracy. The initial ORB-SLAM3 algorithm projection formula is presented in Equation (Equation 1).
(1)$$p_{ij}=f\left(K,R,t,D,P_{ij}\right)$$
where $D=[k_{1},k_{2},p_{1},p_{2},k_{3}]$ represents the aberration parameters. *K*, *R*, and *t* denote the internal parameters and camera position, respectively. $P_{ij}$ and $p_{ij}$ represent the 3D feature points and the feature points calculated using the camera’s intrinsic and extrinsic parameter projections, respectively. In addition, function *f* was the projection function that projected the 3D points to a 2D plane.

Subsequently, the calibrated environment map points were optimized by minimizing the error in the projection equation, as demonstrated in Equation (Equation 2).
(2)$$E=\min\sum_{i}^{n}\sum_{j}^{m}{∥x_{ij}-f\left(R_{j},T_{j},K,D,P_{i}^{W}\right)∥}^{2}$$
where $x_{ij}$ represents the feature points detected in the image. The optimization of the camera’s position and the constructed map points was performed by minimizing the energy function.

In order to optimize the process, calibration plates were fixedly placed in an unknown calibration environment. This was accomplished by affixing a sample plate with a known distance between two points at a fixed location in the surrounding environment, such as the calibration plate depicted in Figure 2.

Subsequently, a constraint component for the distance between known points was added to the optimization function. Specifically, the distances between corner points and neighboring corner points on the squares in the calibration plate were used. This approach aims to obtain more accurate map point data.
(3)$$D={\left|\sqrt{\left(x_{i}-x_{k}\right)+\left(y_{i}-y_{k}\right)+\left(z_{i}-z_{k}\right)}-L_{ik}\right|}^{2}$$

As illustrated in Equation (Equation 3), the values of $x_{i}$ and $x_{k}$ were calculated in order to reduce the number of 3D points representing the corner points on the calibration plate. The distance $L_{ik}$ between the corner points *i* and *k* on the calibration plate was then added to the difference *D*. As this difference was reduced, the map point error was also reduced. Finally, the fixed constraint optimization condition was incorporated into the final optimization function of the algorithm, as illustrated in Equation (Equation 4), with the objective of reducing the map point error.
(4)$$E^{\prime }=\min\sum_{i}^{n}\sum_{j}^{m}∥x_{ij}-f\left(R_{j},T_{j},K,D,P_{i}^{W}\right){∥}^{2}+\alpha \sum_{i}^{n}\sum_{k}^{n-1}{∥D∥}^{2}$$
where the energy function, *E*, was derived from the previous calculation and the constraints, *D*. This function was combined with the newly constructed energy function, $E^{\prime }$ . The goal was to minimize $E^{\prime }$ to reduce the map point error. This approach aimed to enhance the accuracy of the subsequent multi-camera calibration.

### 3.2. Calibration Module

The calibration module accepted constructed multiple map data as input. In this case, the multiple cameras were simplified to two cameras, so the input was the map data constructed by two cameras. Through the calibration module, the output was the initial extrinsic change relationship between the cameras. The specific steps are delineated below.

Matching of similar keyframesFirstly, the keyframe database of the two map data was subjected to keyframe similarity detection using a BOW (bag of words) [24] modeling approach, as illustrated in Figure 3. The keyframes with high similarity in the keyframe database were then selected.Matching of map feature pointsSecondly, among the screened similar keyframes, feature point matching was performed according to the similarity descriptor. Subsequently, a certain number of feature matching points were selected to calculate the extrinsic. To the best of our knowledge, we were the first to propose the selection of feature matching points based on the distance of the feature matching points to compute the extrinsic. This was due to the fact that the effects of feature matching points at different distances on the computation of rotation and translation vectors in the multi-camera epiphenomenon were not uniform. In practice, distant feature matching points played a more significant role in estimating the rotation vectors in the extrinsic, while close feature matching points play a more significant role in estimating the translation vectors in the extrinsic. Then, the selection of feature matching points based on the distance between them could result in the generation of a uniform distribution of feature matching points, which could then be extended to a larger spatial range. This approach could enhance the accuracy of calculating the extrinsic. In our approach, we first set the distant feature matching point threshold and the near matching point threshold. The feature matching points were then classified according to the two thresholds into a far distance feature matching point set and a near distance feature matching point set, and the rest of the feature matching points were classified into a medium distance feature matching point set. The feature matching points with different distances and homogeneous distributions in the three point sets were selected for the subsequent calculation of the multi-camera extrinsic.
(5)$$P_{K}^{A}=C_{B}^{A}\cdot P_{K}^{B},C_{B}^{A}=\left[\begin{array}{cc} R & t\\ 0 & 1 \end{array}\right]$$Calculate the extrinsic parameter transformation relationFinally, based on the feature matches selected in the previous step, the transformation relationship of the multi-camera extrinsic was calculated. As shown in Equation (Equation 5), where $P_{K}^{A}$ and $P_{K}^{B}$ represented the feature matching points in the keyframe K with high similarity, and $C_{B}^{A}$ represents the extrinsic between cameras A and B. According to the equation relationship, the equation was enumerated and the extrinsic $C_{B}^{A}$ was computed. The computed multi-camera extrinsic was set as the initial extrinsic.

### 3.3. Optimize Module

Once the initial extrinsic had been calculated using the calibration module, it was then used to expand the feature matching points. When a sufficient number of feature matching points had been expanded, the extrinsic module was optimized. The optimization part of this paper used bundle adjustment [20], as shown in Figure 4. In the event that the feature points in the two maps corresponded successfully, the corresponding successful map feature points were utilized to project the corresponding successful map points in the first map to the image surface where the other map points were located. Thereafter, the error between the projected map points and the original points in the other map was computed. The error was then minimized through optimization to refine the output of the initial extrinsic transformation relationship obtained from the estimation. The specific formula is shown in Equation (Equation 6).
(6)$$C_{B}^{A}=\mathrm{argmin}\frac{1}{2}\sum_{i=1}^{n}∥u_{i}-C_{B}^{A}\cdot P^{B}∥$$
where $C_{B}^{A}$ represented the extrinsic transformation relationship between camera *A* and camera *B*. $u_{i}$ denotes the image plane feature points in the map constructed by camera A, while $P^{B}$ signifies the 3D feature points in the map constructed by camera *B*. The optimization was achieved by iteratively seeking to minimize the projection error.

The optimization component of the aforementioned procedure enabled the derivation of the multi-camera extrinsic transformation relation. When the extrinsic transformation relation was generalized to more than two cameras, it was based on determining the main camera and then obtaining the extrinsic transformation relation between the auxiliary cameras and the primary camera in turn. The extrinsic reference transformation relationship between the auxiliary cameras was obtained by using the main camera as an intermediate reference. This method derived the extrinsic reference transformation relationship between each auxiliary camera.
(7)$$C_{D}^{B}=C_{A}^{B}\cdot C_{D}^{A},C_{C}^{B}=C_{A}^{B}\cdot C_{C}^{A},C_{D}^{C}=C_{A}^{C}\cdot C_{D}^{A}$$

It could be determined that camera *A* was the primary camera among cameras *A*, *B*, *C*, and *D*. The extrinsic transformation relationship of the secondary camera with respect to the primary camera was then represented by $C_{B}^{A}$, $C_{C}^{A}$, and $C_{D}^{A}$, respectively. The extrinsic transformation relationship between the auxiliary cameras *B*, *C*, and *D* could be calculated using Equation (Equation 7).

## 4. Results

The objective of this experiment is to run the program under Ubuntu 20.04, subsequently analyze the results, and present the results of calibration experiments conducted in the non-overlapping region between multiple cameras. In addition, the reliability of the algorithms is verified by running the program in the monocular camera and the depth camera, respectively.

Firstly, the intrinsic parameters of the camera are determined through the internal reference calibration program utilized for the camera in question. In this instance, the Realsense D435i (Realsense D435i official website: https://www.intelrealsense.com/zh-hans/depth-camera-d435i/ accessed on 28 June 2024) is calibrated for its intrinsic parameters, including focal length, image principal point coordinates, and so forth. The camera is depicted in Figure 5.

The intrinsic parameters $K_{1}$ and $K_{2}$ of each camera were calculating using Equations (Equation 8) and (Equation 9). In the intrinsic reference *K*, the focal lengths $f_{x}$ and $f_{y}$ represent the focal length in the intrinsic reference, while the coordinates of the pixel principal points $c_{x}$ and $c_{y}$ are used to define the intrinsic reference. The multi-camera system is then subjected to experimentation in a virtual environment, namely Gazebo, as well as in a real environment. The objective of this experimentation is to verify the reliability and robustness of the algorithm.
(8)$$K_{1}=\left[\begin{array}{ccc} f_{x} & 0 & c_{x}\\ 0 & f_{y} & c_{y}\\ 0 & 0 & 1 \end{array}\right]=\left[\begin{array}{ccc} 909.72 & 0 & 646.79\\ 0 & 909.08 & 376.31\\ 0 & 0 & 1 \end{array}\right]$$
(9)$$K_{2}=\left[\begin{array}{ccc} f_{x} & 0 & c_{x}\\ 0 & f_{y} & c_{y}\\ 0 & 0 & 1 \end{array}\right]=\left[\begin{array}{ccc} 911.19 & 0 & 651.14\\ 0 & 910.97 & 353.62\\ 0 & 0 & 1 \end{array}\right]$$

### 4.1. Running the Calibration Algorithm in a Simulation Environment

Firstly, the calibration environment and the cart are constructed within a virtual environment. Additionally, one camera is positioned in the front and rear of the cart, as illustrated in Figure 6, which constitutes a multi-camera device, thereby enabling the calibration to be carried out. The front and rear cameras utilized are monocular cameras that capture images at a frequency of 30 Hz, with a resolution of 640×480 .

By navigating the trolley in the virtual calibration environment for closed-loop motion and executing the calibration algorithm, the system is able to perform the calibration of extrinsic camera parameters. The motion trajectory of each camera is shown in Figure 7. The final program outputs the transformation relations of the extrinsic parameters of the front and rear cameras, including the rotation vectors of yaw, pitch, roll, and translations on the *x*, *y*, and *z* axes, for a total of six parameters.

In Table 1, our algorithm is shown against the external parameters of the cammap calibrated multi-camera system and the ground truth of the multi-camera system. We follow up with several experiments to verify the robustness of the algorithm. The variance of each of the rotation and translation vectors in the extrinsic parameter is calculated. The robustness of the algorithm is verified by comparing the variance, which is calculated using Equation (Equation 10). In this equation, $x_{i}$ represents the result of the parameter in the rotation vector or translation vector, E(X) represents the ground truth of the parameter in the rotation vector or translation vector, and D(X) represents the variance of the rotation or translation vector. Table 2 illustrates that the variance of our algorithm in calculating the rotation vectors in the extrinsic parameter is considerably reduced, and it exhibits enhanced robustness in the rotation vectors. This is due to the fact that the feature matching points that are far away in the selection of feature matching points have a greater influence on the rotation vectors in the extrinsic parameter and play a greater role in calculating the rotation vectors in the extrinsic parameter. As illustrated in Table 2, the variance of the computed translation vectors is greater than that of the overall extrinsic parameter, yet the latter is more stable. This is due to the fact that, when selecting feature matching points, we chose far, medium, and close feature matching points, which improved the stability of the rotational vectors more and decreased the stability of the translational vectors slightly. However, feature matching points with varying distances from one another were selected. The use of these points resulted in a spatially uniform distribution, which served to minimize the variance of the overall extrinsic parameter and enhance the stability of the overall extrinsic parameter.

The experimental results of the calibrations performed in the virtual environment are completed with a visual display, as shown in Figure 8, where the relative positions of the front and rear cameras are fully represented in the ROS environment using the coordinate axes.
(10)$$DX=\sum_{i=1}^{n}{\left(x_{i}-EX\right)}^{2}p_{i}$$

### 4.2. Running Calibration Algorithms in a Real Environment

In order to run the calibration algorithm in a real environment, it is first necessary to construct a set of multi-camera equipment. The multi-camera equipment used in this paper is comprised of the front and rear Realsense d435i depth cameras assembled on a fixed bracket, which collectively constitute a non-overlapping multi-camera system, as illustrated in Figure 5.

With the multi-camera device, multi-camera extrinsic reference calibration can be accomplished by extracting feature points in the surrounding calibration environment, as shown in Figure 9, and executing appropriate closed-loop motions, as shown in Figure 10. In the real environment experiment, the front camera is fixed in the front, and the rear camera is rotated from reverse to sideways in a gradual manner to perform multiple calibration experiments.

Table 3 and Table 4 compare the calibration results of the extrinsic of the back camera of the multi-camera rotated from the rear direction to the side direction with the real value of the extrinsic parameter in a real environment. In order to facilitate the presentation of experimental results, we have developed a program for displaying extrinsic parameter results according to the ROS platform. This program displays the multi-camera extrinsic parameter from the front and back cameras for a front and a rear rotated to a front and a side transformed to the coordinate axes, as shown in Figure 11.

## 5. Discussion

In scenarios where multiple cameras have minimal or no overlap in their fields of view, traditional calibration methods cannot achieve accurate calibration between the cameras. Our proposed method provides a solution to the aforementioned problem and is capable of accurately calibrating camera parameters in the case of multiple cameras without overlapping parts. The method is applicable to a wide range of camera types, including monocular and binocular cameras.

In the context of computing multi-camera epiphenomena via feature matching points, previous methods predominantly employed a random selection of feature matching points. The method of randomly selecting feature matching points failed to fully leverage the inherent nature and spatial range distribution of these points. Our proposed method addresses the aforementioned issue by dividing the feature matching points according to their distances. It then selects feature matching points with varying distances and subsequently forms uniformly distributed feature matching points. This approach enhances the precision in calculating multi-camera extrinsic parameters.

Nevertheless, the calibration process remains subject to certain constraints. Prior to executing our proposed algorithm to calibrate the extrinsic of the camera, it is first necessary to calibrate the intrinsic of each camera. This is done in order to ensure the normal execution of our proposed calibration algorithm. Moreover, in order for a multi-camera system to be calibrated, the two cameras must be able to capture the same image. This is because the feature point matching requirement can only be met when the same image is captured by both cameras.

## 6. Conclusions

We propose a calibration method for multi-cameras without overlapping parts that makes use of 3D sparse maps of natural environments. The 3D sparse map of the calibration environment is constructed separately by multiple cameras. Key frames with high similarity are filtered out based on the bag-of-words model. Efficient pairs of feature points are then selected in the frames as a way to run the online calibration algorithm. Finally, the transformation relationship of the extrinsic parameters of the auxiliary camera in the multiple cameras with respect to the main camera is computed to obtain the extrinsic of the multi-camera system. Experiments have been conducted in both virtual and real environments, with reliable experimental results obtained in real environments in both cases where the multi-camera setup is rotated from one front to one back to one front and one side.

The multi-camera extrinsic reference results obtained by the method in this paper can be used to achieve a wider field of view when performing downstream tasks in the field of computer vision, such as semantic segmentation. Utilizing a multi-camera system enables better environmental perception and task accomplishment. In the field of robotics, the multi-camera extrinsic reference results can be utilized to more accurately deploy multi-camera devices in robotic systems for tasks such as the navigation and mapping of unknown environments.

However, there are some limitations when using the methods in this paper for calibration. To calibrate the extrinsic parameters of multiple cameras using this method, the intrinsic parameters of each camera must be known, which is a prerequisite. Additionally, in areas with insufficient feature texture information, the accuracy of feature point matching may degrade due to a lack of features. Despite these challenges, it is essential to persevere and address these issues.

In the future, we will focus on designing a system for the simultaneous calibration of intrinsic and extrinsic parameters of multiple cameras that is highly automated. In scenarios with insufficient point features, we will improve calibration accuracy in challenging environments by combining point and line features to better utilize various features in the natural environment. 

## Figures and Tables

**Figure 1 sensors-24-05228-f001:**
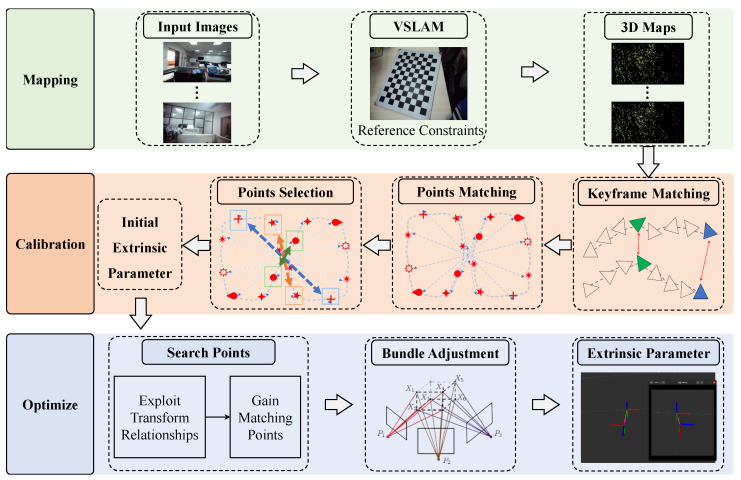
The proposed multi-camera calibration algorithm consisting of mapping, calibration, and subsequent optimization module. The calibration process was carried out sequentially by the order of arrows.

**Figure 2 sensors-24-05228-f002:**
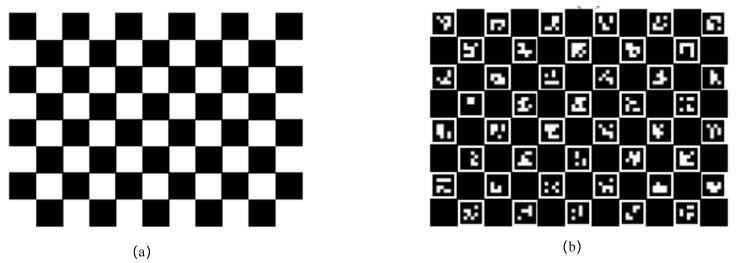
The calibration plate was placed in a calibration environment where the distances between corner points in the calibration plate were known, improving the accuracy of the mapping by the known distances of the corner points. (**a**,**b**) showed two ways in which the features were constructed in two types of calibration plates.

**Figure 3 sensors-24-05228-f003:**
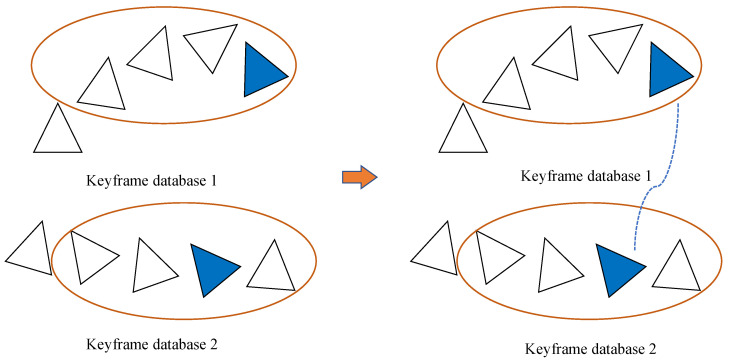
A feature vector was constructed for each keyframe. Using the distances of the feature vectors, keyframes with high similarity were selected. The blue triangles in the figure can then be considered highly similar keyframes.

**Figure 4 sensors-24-05228-f004:**
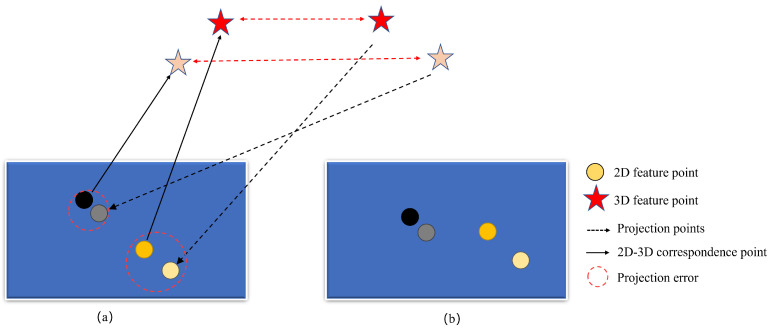
Illustration of bundle adjustment. The reprojection error method is demonstrated by the successful projection of the 3D map points to the image plane, where the original map points reside. (**a**,**b**) represent different image planes where the feature point matches are located, respectively. Dark and light-colored stars connected by red arrows represent matching point pairs at different distances, respectively. Black and dark yellow circles represent original points, and gray and light yellow circles represent projected points. The minimization of the error enables the optimization of the extrinsic transformation relation.

**Figure 5 sensors-24-05228-f005:**
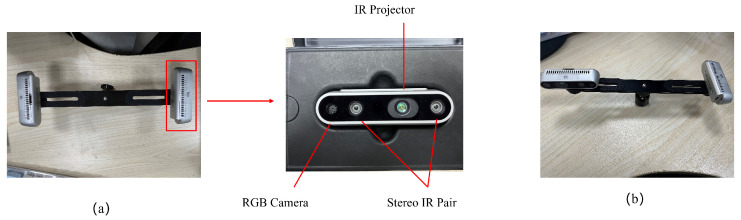
The Realsense D435i camera, which forms part of the multi-camera system, is equipped with a binocular camera as well as a depth camera that employs infrared emission. (**a**,**b**) represent the Realsense D435i multi-camera systems with different extrinsic references, respectively.

**Figure 6 sensors-24-05228-f006:**
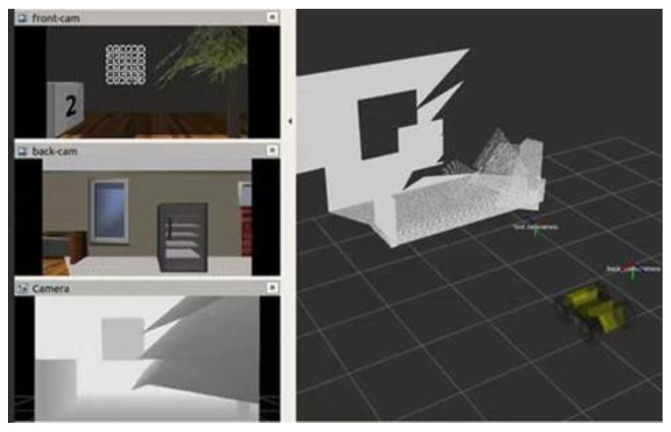
Virtual calibration environment. The modeling of bedrooms, furniture, etc., in Gazebo builds the structure of the environment, as well as a four-wheeled cart equipped with front and rear cameras to form a multi-camera system to ensure the operation in the virtual environment.

**Figure 7 sensors-24-05228-f007:**
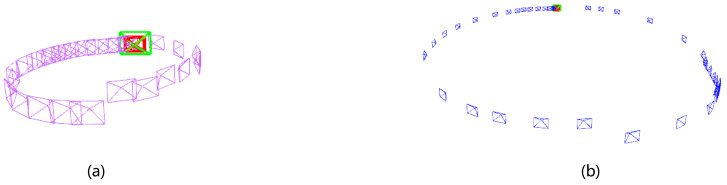
The camera trajectory. Where (**a**) shows the trajectory of the primary camera, where the purple box represents the trajectory and the green color represents the location at that time. And (**b**) shows the trajectory of the secondary camera, where the blue color represents the trajectory and the green color represents the location. Both camera trajectory are performing closed-loop motions to execute the algorithm.

**Figure 8 sensors-24-05228-f008:**
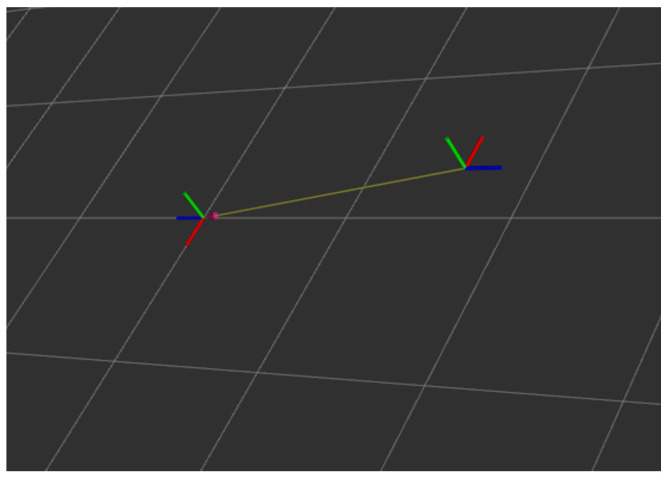
The result of the virtual environment camera extrinsic reference. The blue axes indicate the orientation of the front and rear cameras, while the red and green axes indicate the horizontal and vertical orientation axes of the camera, respectively.

**Figure 9 sensors-24-05228-f009:**
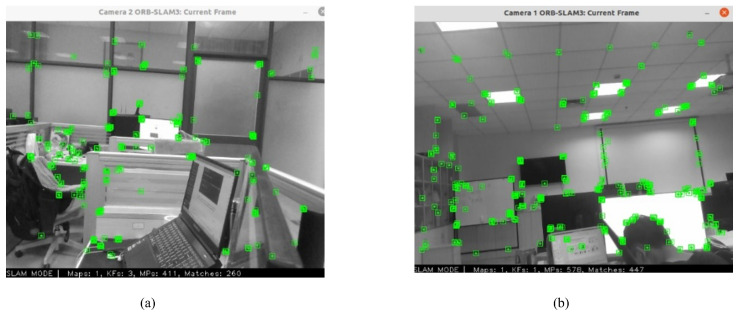
Both the front and rear cameras are capable of extracting feature point images: (**a**) screenshot of the primary camera extracting the sphere feature points in the captured image; (**b**) screenshot of the secondary camera extracting the sphere feature points in the captured image.

**Figure 10 sensors-24-05228-f010:**
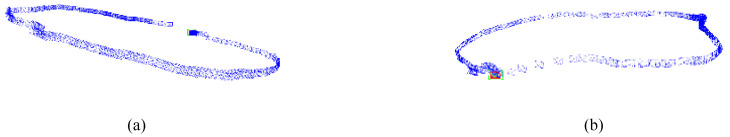
(**a**) shows the motion trajectory of the front camera, (**b**) shows the motion trajectory of the rear camera, and the blue color in both figures indicates the motion trajectory of the camera. Additionally, the front and back cameras complete a closed-loop motion to ensure the operation of the online calibration algorithm.

**Figure 11 sensors-24-05228-f011:**
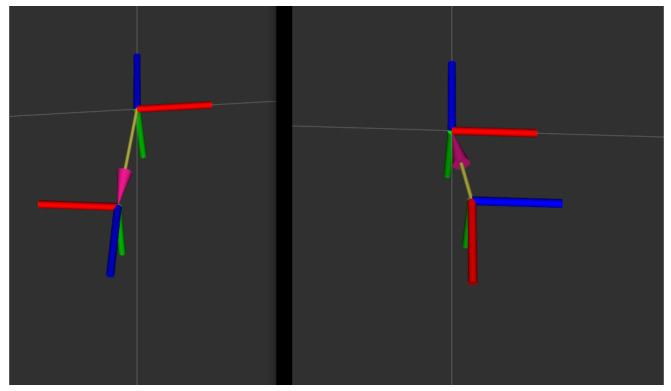
The front and back cameras constitute a demonstration diagram of the extrinsic reference transformation of the multi-camera device. As seed in the figure, the camera orientation indicated by the blue axis is rotated from a front and a rear on the left side to a front and a side on the right side.

**Table 1 sensors-24-05228-t001:** Comparison of the experimental results obtained by the algorithm of this paper with those obtained by the Cammap algorithm in a virtual environment. In the parameters with downward arrows, it indicates that the smaller the data is, the more accurate it is.

Extrinsic	Rotation (°)	Inaccuracies (Rota) ↓	Translation (m)	Inaccuracies (Trans) ↓
**Ground truth**	[−180, 0, −180]	—	[0, 0.356, −1.932]	—
**Cammap**	[−181.189, −0.514, −179.790]	[1.890, 0.514, **0.210**]	[0.087, 0.297, −1.878]	[0.087, **0.059, 0.054**]
**Ours**	[−179.050, −0.430, −179.585]	[**0.950, 0.430**, 0.415]	[0.025, 0.261, −2.030]	[**0.025**, 0.095, 0.098]

Bolded numbers represent smaller errors.

**Table 2 sensors-24-05228-t002:** Stability comparison of this paper’s algorithm with the extrinsic parameters computed from cammap results in a virtual environment. In the parameters with downward arrows, it indicates that the smaller the data is, the more accurate it is.

Variance	Rotation (°) ↓	Translation (m) ↓	Overall ↓
**Cammap**	0.631	**0.0653**	0.6993
**Ours**	**0.590**	0.0726	**0.6626**

Bolded numbers represent smaller errors.

**Table 3 sensors-24-05228-t003:** Front and a rear cameras constitute the extrinsic parameters of a multi-camera device.

Extrinsic	Front and Rear Cameras	Ground Truth
Rotation (°)	[−177.399, −4.1353, −177.957]	[−180, 0, −180]
Translation (m)	[−0.0288, −0.05166, −0.28815]	[0, 0, −0.250]

**Table 4 sensors-24-05228-t004:** Front and side cameras constitute the extrinsic parameters of a multi-camera device.

Extrinsic	Front and Side Cameras	Ground Truth
Rotation (°)	[−100.476, 87.9655, −100.297]	[−90, 90, −90]
Translation (m)	[0.0439, −0.0213, −0.198614]	[0, 0, −0.250]

## Data Availability

Data are contained within the article.

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
