# Peer review of "NMC3D: Non-Overlapping Multi-Camera Calibration Based on Sparse 3D Map"

_sensors, 2024, doi:10.3390/s24165228_

Round 1

Reviewer 1 Report

Comments and Suggestions for Authors

The reviewer comments are in the pdf file

Comments on the Quality of English Language

Reviewer 2 Report

Comments and Suggestions for Authors

In this paper, a calibration method for multi-camera systems without overlapping part is proposed. The method is interesting. However, there are still some issues that need improvement.

(1) In section 3, how to obtain the 3D maps from the input images and what is the function of a chessboard board, shown in Figure 1. More detailed introduction should be given.

(2) In equation 4, are the corner points on the calibration plate automatically selected? How to distinguish between environmental feature points and checkerboard feature points?

(3) The application scenarios of the method in this article should be clearly defined. Under what circumstances can only this camera setting be used, i.e. the multi-camera systems without overlapping parts.

(4) Suggest adding more camera deployment examples for method validation.

Reviewer 3 Report

Comments and Suggestions for Authors

This paper summarizes the research status of multi-camera system calibration in recent years, and points out that the calibration accuracy of multi-camera systems under non-overlapping views still has a lot of room for improvement. By screening out evenly distributed matching points among similar feature points in the constructed multiple environmental maps, the calibration algorithm is used to obtain the accurate conversion relationship and external parameters between cameras, thereby improving the camera calibration accuracy. However, in the virtual environment constructed in the experimental phase, only a set of experimental comparison results between the algorithm in this article and the Cammap algorithm cannot fully reflect the superiority of the algorithm. In addition, the virtual environment and the real environment map are single, which cannot well reflect the generalization of the algorithm.

Comments on the Quality of English Language

Minor editing of English language is required.

Round 2

Reviewer 1 Report

Comments and Suggestions for Authors

The authors realized all corrections and improvements suggested and required.

Comments on the Quality of English Language

minor

Reviewer 2 Report

Comments and Suggestions for Authors

there is no more comments.